# New Cardiovascular Risk Biomarkers in Rheumatoid Arthritis: Implications and Clinical Utility—A Narrative Review

**DOI:** 10.3390/biomedicines13040870

**Published:** 2025-04-03

**Authors:** Anna Pamies, Joan-Carles Vallvé, Silvia Paredes

**Affiliations:** 1Secció de Reumatologia, Hospital de Tortosa Verge de la Cinta, 43500 Tortosa, Catalonia, Spain; apamies.ebre.ics@gencat.cat; 2Unitat de Recerca en Lípids i Arterioesclerosi, Universitat Rovira i Virgili, 43204 Reus, Catalonia, Spain; silvia.paredes@salutsantjoan.cat; 3Institut Investigació Sanitaria Pere Virgili, 43204 Reus, Catalonia, Spain; 4Centro de Investigación Biomédica en Red de Diabetes y Enfermedades Metabólicas Asociadas (CIBERDEM), Instituto de Salud Carlos III (ISCIII), 28029 Madrid, Spain; 5Secció de Reumatologia, Hospital Universitari Sant Joan de Reus, 43204 Reus, Catalonia, Spain

**Keywords:** cardiovascular disease, rheumatoid arthritis, interleukin-32, Dickkopf-1, galectin-3, catestatin, fetuin-A

## Abstract

Rheumatoid arthritis (RA) is a chronic autoimmune disease that not only causes joint inflammation but also significantly increases the risk of cardiovascular disease (CVD), leading to a higher morbidity and mortality. RA patients face an accelerated progression of atherosclerosis, attributed to both traditional cardiovascular risk factors and systemic inflammation. This review focuses on emerging biomarkers for cardiovascular risk assessment in RA, aiming to enhance early detection and treatment strategies. Specifically, we examine the roles of interleukin-32 (IL-32), Dickkopf-1 (DKK-1), galectin-3 (Gal-3), catestatin (CST), and fetuin-A (Fet-A) as potential markers for CVD in this patient population. IL-32, a proinflammatory cytokine, is elevated in RA patients and plays a significant role in inflammation and endothelial dysfunction, both of which contribute to atherosclerosis. DKK-1, a Wnt signaling pathway inhibitor, has been associated with both synovial inflammation and the development of atherosclerotic plaques. Elevated DKK-1 levels have been linked to an increased CV mortality and could serve as a marker for CVD progression in RA. Gal-3 is involved in immune modulation and fibrosis, with elevated levels in RA patients correlating with disease activity and cardiovascular outcomes. Catestatin, a peptide derived from chromogranin A, has protective anti-inflammatory and antioxidative properties, though its role in RA-related CVD remains under investigation. Finally, Fet-A, a glycoprotein involved in vascular calcification, shows potential as a biomarker for CV events in RA, though data on its role remain conflicting. These biomarkers provide deeper insights into the pathophysiology of RA and its cardiovascular comorbidities. Although some biomarkers show promise in improving CV risk stratification, further large-scale studies are required to validate their clinical utility. Currently, these biomarkers are in the research phase and are not yet implemented in standard care. Identifying and incorporating these biomarkers into routine clinical practice could lead to the better management of cardiovascular risk in RA patients, thus improving outcomes in this high-risk population. This review highlights the importance of continued research to establish reliable biomarkers that can aid in both diagnosis and the development of targeted therapies for cardiovascular complications in RA.

## 1. Introduction

Rheumatoid arthritis (RA) is a chronic, autoimmune, and systemic disease that primarily affects synovial joints. It has a prevalence of approximately 1% in the general population and is more common in women. In most patients, without appropriate treatment, the disease follows a progressive course, leading to joint destruction, the impairment of physical function, and a reduced quality of life. Additionally, RA patients exhibit an increased mortality compared to the general population [1,2].

Cardiovascular disease (CVD) is the leading cause of morbidity and mortality in these patients, with acute myocardial infarction (AMI) being the most frequent event [3]. It is estimated that the risk of CVD is approximately 50% higher in RA patients compared to the general population [3,4,5]. Consequently, RA is considered an independent risk factor for developing CVD, with a prevalence similar to that observed in patients with type 2 diabetes mellitus (T2DM) [6] and a standardized mortality rate for AMI that can reach up to 3.82 in certain populations [7]. To assess this elevated CV risk, surrogate markers such as carotid intima-media thickness (cIMT) and arterial stiffness have been utilized. Recent studies highlight complex interactions between disease activity, patient age, and sex, with systolic blood pressure and body mass index (BMI) identified as potential mediators of these associations [8,9].

In RA, synovial fibroblasts and macrophages produce proinflammatory cytokines such as tumor necrosis factor-alpha (TNF-α), interleukin-6 (IL-6), and IL-1, while immune cell infiltration leads to synovial pannus formation, contributing to joint destruction [10,11]. This chronic inflammation shares key features with atherosclerosis [12], where elevated cytokine levels, particularly TNF-α and IL-6, promote endothelial dysfunction, a key initial step in atherosclerosis pathophysiology [13]. Chronic inflammation has been proposed as a contributor to the initiation and progression of accelerated atherosclerosis in RA patients [14], further linking RA with an increased cardiovascular risk. Additionally, persistent inflammation in RA reduces the antioxidant capacity by depleting antioxidant vitamins [15], exacerbating oxidative stress. Genetic variability also plays a significant role, influencing the intensity of the inflammatory response and contributing to disease heterogeneity [16].

Both traditional cardiovascular (CV) risk factors—such as age, sex, smoking, type 2 diabetes mellitus (T2DM), and hypertension—and systemic inflammation, contribute to the elevated CV risk in RA patients [10]. However, these factors alone do not fully account for the accelerated atherosclerosis observed in this population. While some studies report a higher prevalence of hypercholesterolemia, hypertension, smoking, and obesity in RA patients [17,18], others suggest similar rates to the general population [19]. Advanced lipid profiling, including the analysis of lipoprotein subfraction size and number, has provided more clinically relevant insights [20]. Additionally, hypertension and dyslipidemia are often underdiagnosed and undertreated in RA, with metabolic syndrome further exacerbating CV risk [21,22].

Despite limited data on primary prevention strategies in RA, population-based studies have shown that statin use is associated with a reduced CV risk [23,24]. A prospective interventional study demonstrated that managing traditional CV risk factors led to a 50% reduction in subclinical atherosclerosis progression and a lower incidence of first CV events in RA patients following a treat-to-target approach [25]. Moreover, disease-modifying antirheumatic drugs (DMARDs), which effectively control RA disease activity, have also been linked to a decreased CV risk, highlighting the interplay between inflammation control and cardiovascular protection [26,27].

As outlined above, CVD represents a significant comorbidity in RA patients, underscoring the importance of early diagnosis for the development and implementation of effective therapeutic strategies. In this context, identifying novel biomarkers that enable early diagnosis and improved CV risk stratification in RA patients is a priority. In addition to classical biomarkers, newly identified biomarkers such as inflammatory markers characterized by nuclear magnetic resonance [28] or microRNAs [29,30,31,32] enhance the precision of CV risk assessment in RA patients. These biomarkers not only provide a greater insight into disease pathophysiology but also serve as critical tools for identifying patients at high risk for CV events. In recent years, various molecules have been proposed as emerging CV risk biomarkers. Among these candidates, this narrative review focuses on five specific biomarkers: interleukin-32 (IL-32), Dickkopf-1 (DKK1), galectin-3 (Gal3), catestatin (CTS), and fetuin-A (Fet-A). This review examines the current evidence regarding the relationship between these biomarkers and CVD in RA patients, exploring their mechanisms of action and potential utility as clinical CV risk biomarkers in this population.

## 2. Methods

For this narrative review, we conducted a comprehensive search of the PubMed, MEDLINE, and ScienceDirect databases. Our focus centered on five biomarkers (IL-32, DKK-1, Gal-3, CST, and Fet-A) and their associations with CV disease and CV risk in RA. The search strategy involved using each biomarker’s name as a primary keyword in combination with “CV disease” and “CV risk”, as well as additional searches combining each biomarker’s name with “rheumatoid arthritis” to ensure specificity to RA-related CV risk. To further refine the results, we conducted additional searches by cross-referencing biomarker-related cardiovascular terms with “rheumatoid arthritis”. The search was restricted to articles published from the year 2000 onward. An initial screening identified 1908 potentially relevant records, of which 177 met our inclusion criteria following an abstract review.

## 3. Interleukin-32

IL-32 is a proinflammatory cytokine produced by both immune and non-immune cells in response to various stimuli. It exists in multiple isoforms, with IL-32γ being the most active [33]. IL-32 is involved in diverse immune responses, inflammation, and the control of infectious diseases, playing a significant role in CV health, particularly in the context of chronic inflammation and atherosclerosis. It participates in the modulation of key processes in CVD, including inflammatory pathways, endothelial cell function, and lipid metabolism [34] (Figure 1).

### 3.1. Clinical Evidence on the Role of IL-32 in CVD

There is extensive evidence indicating that IL-32 levels are elevated in CVD patients. For instance, increased IL-32 levels have been observed both in plasma and within coronary artery lesions in patients with ischemic heart disease. Moreover, IL-32 levels progressively increase in patients with stable angina, unstable angina, and AMI, suggesting a correlation with disease severity [35]. Furthermore, studies such as that by Xuan et al. in 2017 [36] have demonstrated that elevated IL-32 levels have significant prognostic value, as they are associated with a markedly increased CV risk. Patients with heart failure following AMI exhibited elevated IL-32 levels, which correlated with pro B-type natriuretic peptide (proBNP) levels and a threefold higher risk of cardiac events during follow-up [36].

Additionally, IL-32 levels have been shown to increase after hypoxia/reoxygenation in cardiomyocytes, which is associated with increased oxidative stress, inflammation, and cellular apoptosis. This suggests that IL-32 may play a role in myocardial ischemia-reperfusion injury [37]. Moreover, a recently identified single nucleotide polymorphism (SNP) in the IL-32 gene has been linked to increased CV mortality [38], suggesting that genetic variants in this gene could have a significant impact on CV risk prediction and prognosis. Collectively, these findings highlight IL-32 as an emerging biomarker associated with increased CV risk.

### 3.2. Clinical Evidence on the Role of IL-32 in RA

Clinical evidence suggests that IL-32 plays a crucial role in RA and may serve as a biomarker of disease activity and manifestations. IL-32 is expressed in various tissues and cells, including lymphoid tissue, leukocytes, stimulated epithelial cells, and synovial fibroblasts. Moreover, IL-32 levels correlate with the intensity of inflammation in synovial tissue biopsies from RA patients [39]. Interestingly, IL-32γ has been identified as the isoform most strongly associated with inflammation in RA. Increased levels of IL-32γ in synovial biopsies and IL-32 in synovial fibroblasts have been demonstrated in RA patients compared to those with osteoarthritis, suggesting that the modulation of IL-32 isoforms could be a strategy for controlling inflammation [40,41]. Animal model studies have shown that an intra-articular injection of IL-32 induces arthritis and joint damage, with a correlation between synovial inflammation severity and IL-32 levels, indicating a potential pathogenic role of IL-32 in RA [42].

### 3.3. Mechanisms of Action of IL-32

IL-32 modulates key inflammatory pathways by inducing the production of proinflammatory cytokines such as IL-6, IL-1, and particularly TNF-α [43]. This mechanism is crucial for IL-32-mediated joint damage, as studies in TNF-α-deficient mice have shown no inflammation despite IL-32 overexpression [42,44]. Notably, IL-32γ shows the strongest correlation with TNF-α levels, whereas IL-32β does not correlate with TNF-α or other proinflammatory cytokines [40]. This suggests an autoinflammatory loop between IL-32γ and TNF-α, where both cytokines enhance each other’s activity. Interrupting this loop with anti-TNFα therapies has been shown to reduce IL-32 overexpression, which may contribute to the potent anti-inflammatory effects of these treatments [45]. Additionally, it has been described that signaling via Syk/protein kinase C delta (PKCδ)/Janus kinase 3 upregulates the IL-32 production in synovial fibroblasts stimulated with TNF-α [46]. Furthermore, evidence suggests that IL-32, particularly IL-32γ, induces dendritic cell activation and the production of IL-12 and IL-6, promoting T cell polarization towards Th1 and Th17 lineages [47].

Beyond its role in joint inflammation, IL-32 also affects key mechanisms related to atherogenesis. This cytokine has been shown to enhance lipid deposition and attenuate cholesterol efflux, one of the most critical anti-atherogenic mechanisms [48]. Furthermore, IL-32 has been implicated in modulating endothelial cell function and serum high-density lipoprotein (HDL) levels [34,35]. There is also evidence supporting IL-32′s involvement in atherosclerosis pathogenesis, as an increased IL-32 expression has been observed in arterial plaques from patients, and studies in IL-32γ transgenic mice have demonstrated early arteriosclerotic damage [49]. Recent studies in healthy populations have associated IL-32 with atherogenic lipoproteins [50] and with monocyte endothelial adhesion mediated by intercellular adhesion molecule-1 (ICAM-1) and vascular cell adhesion molecule-1 (VCAM-1), a crucial step in atherosclerosis pathogenesis. This adhesion is attenuated by the IL-32θA94V mutation in the IL-32θ isoform, which may act as a protective cytokine in atherosclerosis [51].

### 3.4. Evidence Supporting IL-32 as a Biomarker of CVD Risk in RA

Although specific data on the utility of IL-32 levels as a biomarker for CV risk assessment in RA patients are currently lacking, genetic findings suggest its potential involvement. Notably, the influence of a functional SNP in the IL-32 gene (rs4786370) has been investigated, revealing that RA patients carrying this variant have higher circulating HDL concentrations, potentially indicating a protective effect against CVD [52]. Furthermore, in patients with non-alcoholic fatty liver disease and metabolic dysfunction, IL-32 levels have been associated with poor blood pressure control, the primary CV risk factor [53]. However, its association with CV risk extends beyond RA. Elevated IL-32 levels have also been observed in other chronic inflammatory conditions, such as metabolic disorders and HIV infection, which indicates that this cytokine may not be exclusive to RA. For instance, in patients with HIV, increased IL-32 levels have been associated with subclinical markers of atherosclerosis [54,55,56]. Additionally, IL-32 has been implicated in other chronic diseases with increased CV risk, such as chronic obstructive pulmonary disease (COPD) and inflammatory bowel disease (34), reinforcing its broader role in systemic inflammation. Nevertheless, its overexpression in synovial tissue—particularly the IL-32γ isoform—suggests a distinct disease-specific dimension in RA. IL-32’s interplay with TNF-α and other inflammatory mediators is known to drive synovial inflammation and endothelial dysfunction, both of which could further heighten CV risk in this patient population. Future studies should focus on longitudinal evaluations that compare IL-32 levels in RA with those in other inflammatory disorders to determine whether its predictive value is indeed specific to RA-related CV complications or reflects a more generalized inflammatory state.

Overall, the available evidence suggests that IL-32 may serve as a candidate biomarker for CV risk in RA, supported by its involvement in RA pathophysiology, atherosclerosis, and systemic inflammation. In addition, its role in disease mechanisms makes it a promising therapeutic target worthy of further investigation.

## 4. Dickkopf-1

DKK-1 is a multifunctional secreted protein that acts as a natural inhibitor of the Wnt/β-catenin signaling pathway, which is essential for regulating various biological processes, including embryonic development, tissue homeostasis, and tissue regeneration. The Wnt pathway plays a crucial role in bone metabolism, vascular function, cell proliferation, and survival. Consequently, DKK-1 dysregulation has significant implications in multiple pathologies, contributing to cancer progression, immune evasion, bone remodeling, diabetic complications, cognitive function, and CV risk [57] (Figure 2).

### 4.1. Clinical Evidence on the Role of DKK-1 in CVD

DKK-1 plays a crucial role in atherosclerosis and various CVD. Previous studies have documented elevated DKK-1 levels in patients with arteriosclerosis, coronary artery disease (CAD), or carotid plaques [58]. In this regard, high DKK-1 levels have been associated with coronary calcifications and independently predict the presence of atherosclerotic plaques [59]. Moreover, plasma DKK-1 levels have been found to be significantly higher in patients with acute stroke compared to those with stable cerebrovascular disease, and both groups show higher levels than healthy individuals [60]. Further research has confirmed that DKK-1 levels act as an independent marker of stroke [61]. In the context of premature AMI, studies from 2012 linked this event to elevated DKK-1 levels, suggesting that reduced Wnt signaling may be a contributing factor [62]. Additionally, in diseases with a high CV risk, such as T2DM, elevated DKK-1 levels have been observed compared to healthy controls, correlating with endothelial dysfunction and platelet activation [63]. Conversely, diabetic patients with lower baseline DKK-1 levels exhibited better cardiometabolic parameters at baseline [64].

The role of DKK-1 as a predictor of CV mortality has been extensively studied. Elevated DKK-1 levels upon hospital admission have been associated with an increased CV mortality in patients with acute coronary syndrome (ACS) [65]. Similarly, in ischemic stroke, high baseline DKK-1 levels have been linked to a greater mortality [61] and long-term disability [66]. Another piece of evidence linking DKK-1 as a CV risk factor is the improvement of its levels through protective measures. A 2020 study showed that exercise in diabetic patients reduced both cIMT and DKK-1 levels, likely due to decreased blood flow shear stress induced by physical activity [67]. Additionally, treatments such as low-dose aspirin administration and proper diabetes management lower DKK-1 levels [63,64], reinforcing its potential as a therapeutic marker.

However, controversies exist regarding the role of DKK-1 in specific populations. In hemodialysis patients, some studies found no correlation between DKK-1 levels and CV events, arterial stiffness, or mortality [68,69,70], which could be attributed to specific alterations in the Wnt/β-catenin pathway inherent to this population [71]. Additionally, other studies have reported negative associations between plasma DKK-1 levels and vascular calcifications in elderly men, postmenopausal women, and hemodialysis patients [72,73,74]. These findings have been attributed to the potential retention of DKK-1 in calcified plaques, which could explain its low detectable levels in plasma. However, recent Mendelian randomization studies have demonstrated a causal association between genetic predisposition to elevated DKK-1 levels and an increased risk of coronary artery disease and ischemic stroke, further supporting DKK-1’s potential as a therapeutic target for the prevention and treatment of these conditions [75].

### 4.2. Clinical Evidence on the Role of DKK-1 in RA

Several studies have demonstrated that DKK-1 levels are significantly elevated in RA patients. A 2018 meta-analysis confirmed this elevation compared to healthy controls [76]. Moreover, DKK-1 overexpression in RA has been associated with synovial fibroblast migration and enhanced synovial inflammation [77]. In this context, a 2021 study reported that DKK-1 is highly expressed in both the synovial fluid and synovial fibroblasts of RA patients [78]. Additionally, proinflammatory cytokines increase DKK-1 expression, while DKK-1 overexpression further enhances cytokine production, creating a self-perpetuating inflammatory loop that sustains disease progression [78].

The role of DKK-1 in structural damage progression in RA remains controversial. A 2016 study conducted in tightly controlled RA patients found no association between DKK-1 levels and radiographic progression [79]. However, another study from the same year, based on the ESPOIR cohort (early-onset RA patients), found that baseline DKK-1 levels were an independent predictor of structural damage progression at two years [80].

At the genetic level, several studies have explored the relationship between DKK-1 SNPs and structural damage in RA. In the ESPOIR cohort, three SNPs (rs1896368, rs1896367, and rs1528873) were initially associated with radiographic progression and structural damage, with rs1896368 being the most relevant polymorphism [81]. However, subsequent studies failed to replicate these findings [82]. A more recent 2020 study suggested that rs1896367 was linked to a greater structural damage, while rs1896368 might have a protective effect [83].

Thus, DKK-1 plays a multifaceted role in RA, from its contribution to synovial inflammation to its potential involvement in structural damage. The influence of genetic variations in the DKK-1 gene highlights the need for further research into its pathophysiological role in RA and its potential as a biomarker or therapeutic target.

### 4.3. Mechanisms of Action of DKK-1

Multiple mechanisms contribute to the role of DKK-1 as a key biomarker in inflammatory and CV diseases. Platelets have been identified as a cellular source of DKK-1, with this biomarker playing a significant role in endothelial cell activation. Additionally, DKK-1 promotes the release of proinflammatory cytokines associated with endothelial dysfunction and atherosclerosis development [58]. Moreover, DKK-1 is involved in TNF-α-mediated mechanisms, a crucial factor in both RA and platelet-dependent endothelial activation. These processes not only exacerbate endothelial dysfunction but are also linked to vascular calcification and premature CV events [58,62,63]. Furthermore, increased DKK-1 levels associated with atherosclerosis have been correlated with shear stress exerted on endothelial cells, further supporting its role in vascular pathology [84].

### 4.4. Evidence Supporting DKK-1 as a Biomarker of CVD Risk in RA

Although DKK-1 has emerged as a key mediator in the interaction between inflammation, atherosclerosis, and CV risk, its utility as a prognostic biomarker for CVD in RA patients has not yet been thoroughly evaluated. However, some studies provide indirect evidence suggesting its potential relevance. A 2017 study analyzing myocardial perfusion defects in 189 RA patients found a significant association between DKK-1 levels and the presence of these defects [85], supporting its possible role in CV complications in this population. Similarly, a recent study in patients with psoriatic arthritis identified a correlation between DKK-1 levels and cIMT [86], reinforcing its relevance in the context of inflammatory disease. However, a recent study in RA patients found no significant associations between serum DKK-1 levels and traditional CV risk markers, such as cIMT or the presence of carotid plaque [87].

In this context, the lack of robust studies directly assessing the role of DKK-1 as a CV risk biomarker in RA patients presents an opportunity for future research. Further exploration in this area could help establish its prognostic utility and contribute to the clinical management of this population, a goal of undeniable clinical importance.

## 5. Galectin-3

Gal-3 is a β-galactoside-binding lectin, belonging to the family of carbohydrate-binding proteins, and is widely expressed in various cell types, including immune cells, fibroblasts, and endothelial cells. It plays a crucial role in regulating biological and pathological processes, including inflammation, tissue remodeling, apoptosis, and fibrosis [88]. Its involvement in CV health and chronic inflammatory diseases, such as rheumatoid arthritis, positions it as a key regulator in these contexts [89,90] (Figure 3). These characteristics highlight its relevance as a clinical biomarker and as a promising therapeutic target for the development of new treatment strategies in inflammatory and CV diseases.

### 5.1. Clinical Evidence on the Role of Gal-3 in CVD

Gal-3 has been extensively studied for its involvement in various CVDs, with substantial clinical evidence supporting its role as a biomarker and potential therapeutic target. In particular, Gal-3 has been widely investigated in heart failure, where it has been associated with both disease progression and the occurrence of adverse clinical outcomes [91]. In this context, elevated Gal-3 levels in heart failure patients have been linked to a more severe ventricular remodeling, reduced ejection fraction, and increased tissue fibrosis [91,92]. Moreover, its persistent elevation has been identified as a significant predictor of mortality and new-onset heart failure [93], establishing it as a key biomarker for the evaluation and management of this condition.

On the other hand, it has been reported that in patients with metabolic syndrome and T2DM, elevated Gal-3 levels increased the risk of developing heart failure fourfold [94], reinforcing its importance in CV risk stratification. Additionally, recent studies suggest that the use of Gal-3 inhibitory proteins could be a promising strategy to prevent myocardial fibrosis and reduce the incidence of heart failure after myocardial infarction [95]. It has also been observed that older individuals with low Gal-3 levels had a significantly lower CV risk, highlighting its potential as a protective marker in this group [96].

Gal-3 has been extensively studied as a prognostic biomarker in various CVDs. Multiple studies have demonstrated that elevated serum Gal-3 levels are associated with worse CV outcomes and increased mortality across different patient populations. A 2017 meta-analysis including eighteen studies with more than 30,000 subjects found that elevated Gal-3 levels were linked to increased all-cause and CV mortality in both heart failure patients and the general population. This effect persisted even after adjusting for proBNP and renal function, underscoring the robustness of its prognostic utility [97]. Subsequently, a 2020 meta-analysis including 18 studies with 7057 patients confirmed that high Gal-3 levels were associated with all-cause and CV mortality in heart failure patients [98]. However, previous findings presented some controversy. A 2012 study found no significant association between Gal-3 levels and events such as CV death, acute myocardial infarction, or stroke in heart failure patients over 60 years old, a discrepancy attributed to statistical adjustment for proBNP [99]. Furthermore, in this context, a study on obese Chinese adults found an inverse relationship between Gal-3 levels and subclinical atherosclerosis measured by carotid intima-media thickness [100], contrasting with previous evidence. More recently, a 2023 meta-analysis detected an association between elevated Gal-3 levels and the risk of incident heart failure, strengthening the accumulated evidence [101]. Additionally, in stroke patients, high Gal-3 concentrations have been significantly associated with death or disability at three months in a cohort of 3082 patients, reinforcing its relevance as a prognostic biomarker in cerebrovascular events [102]. Gal-3 has also been linked to specific mechanisms such as increased myocardial fibrosis and a higher risk of sudden death [103] and, in hemodialysis patients, elevated Gal-3 levels have been associated with increased mortality, possibly explained by enhanced vascular calcification [104]. Finally, data from patients with T2DM and sleep apnea syndrome showed a positive association between Gal-3 levels and atherosclerosis burden [105,106], findings consistent with the extensive evidence of Gal-3’s role in the progression of this condition [89,107,108].

In conclusion, Gal-3 can be considered a key biomarker in CV diseases, particularly for its ability to predict adverse outcomes and mortality.

### 5.2. Clinical Evidence on the Role of Gal-3 in RA

The data published to date implicate Gal-3 in modulating the immune response in RA, promoting chronic inflammation and joint degradation through the activation of proinflammatory cytokines and the regulation of synovial fibroblasts. In this context, elevated Gal-3 levels have been observed in both the serum and synovial fluid of RA patients compared to healthy controls, with this elevation correlating with inflammatory markers such as C-reactive protein, as well as with synovial fibroblast activation and joint destruction [109]. Additionally, synovial fibroblasts from RA patients, when interacting with cartilage oligomeric matrix protein, exhibited a significant increase in intracellular Gal-3 accumulation, a phenomenon occurring four times more frequently than in fibroblasts from osteoarthritis patients [110]. Furthermore, a 2017 study showed that baseline Gal-3 levels in newly diagnosed RA patients correlated with factors such as smoking, anti-citrullinated peptide antibodies, and the presence of joint erosions, suggesting that Gal-3 may be involved in pathways leading to structural damage [111]. The authors also concluded that Gal-3 levels could be useful for distinguishing patients with undifferentiated RA from those without RA [112]. Additionally, in two independent cohorts of RA patients treated with disease-modifying antirheumatic drugs and/or glucocorticoids, Gal-3 levels were found to be reduced, likely because of treatment. However, this protein maintained a positive correlation with patients’ functional capacity and demonstrated the ability to differentiate between RA and non-RA patients [90].

Moreover, genetic association studies have identified three SNPs in the LGALS3 gene (rs1009977, rs4644, and rs74050921) associated with RA susceptibility [113]. Later, in 2021, it was confirmed in an Asian population that the SNPs rs1009977 and rs3751093 were significantly more frequent in RA patients than in controls, with rs1009977 correlating with specific clinical manifestations of the disease [114].

Finally, Gal-3 has recently been evaluated as a diagnostic biomarker for RA, demonstrating a high diagnostic capacity to identify RA patients compared to controls through receiver operating characteristic (ROC) analysis [115]. Similarly, a 2022 study showed that a Gal-3 inhibitor significantly reduced inflammation and osteoclastogenesis in cell cultures, highlighting its potential as a therapeutic target in this disease [116].

### 5.3. Mechanisms of Action of Gal-3

As previously mentioned, Gal-3 is involved in various cellular processes, playing a significant role in inflammation and fibrosis through different cellular signaling pathways [117]. Its action contributes to a wide range of CV effects that exacerbate these pathologies. In the inflammatory context, Gal-3 has been shown to increase the production of proinflammatory cytokines, such as TNF-α and IL-6, in both synovial fibroblasts and animal models [118,119]. Additionally, Gal-3 promotes inflammation by recruiting monocytes to the arterial wall, enhancing the formation of atheromatous plaques [88]. It has also been reported to contribute to macrophage infiltration and differentiation, further aggravating inflammation and interstitial fibrosis in the context of myocardial infarction [88,120]. Furthermore, Gal-3 has been described to promote cardiomyocyte apoptosis, oxidative stress, and fibrosis through the activation of the NF-κB p65 pathway, leading to cardiac dysfunction [121]. Additionally, Gal-3 interacts with glycans exposed by organelles damaged under oxidative stress, activating cell death signaling cascades in cardiomyocytes and exacerbating CV damage [122]. Finally, Gal-3 has been shown to enhance platelet aggregation and thrombosis through Dectin-1 activation, increasing calcium influx and reactive oxygen species production, thereby contributing to CV complications [123].

### 5.4. Evidence Supporting Gal-3 as a Biomarker of CVD Risk in RA

Gal-3 is emerging as a promising biomarker for assessing CV risk in patients with rheumatoid arthritis, although its specific role as a prognostic biomarker in this context is still under investigation. While the available evidence is preliminary and limited to small cohorts, current data support its potential utility in evaluating CV disease in this population using clinical variables as indirect markers. In this regard, one study found that elevated Gal-3 levels were independently associated with a decreased cardiac output and increased systemic vascular resistance in RA patients compared to a control group. These findings highlight a possible relationship between Gal-3 and hemodynamic alterations in this context [124]. Additionally, in another recent study, although elevated Gal-3 levels were observed in a cohort of RA patients compared to healthy controls, no association was found with myocardial perfusion [125]. On the other hand, more recent research has demonstrated that elevated Gal-3 levels in RA patients are associated with aortic inflammation, as assessed by PET/CT. However, in the same study, no significant associations were found between Gal-3 and coronary calcification or myocardial flow reserve [126].

Overall, Gal-3 shows considerable potential as a biomarker for assessing CV risk in RA patients due to its association with CV markers and its elevated levels in this population. Variations in the reported relationship between Gal-3 and CV risk in patients with RA may be partly explained by its differential utility across disease stages. Gal-3 could be more informative in the early stages of atherosclerosis, when vascular changes are potentially reversible, while its predictive value may decline in more advanced stages of CVD. Additionally, differences in study design, patient selection, and assay methodologies likely contribute to the observed inconsistencies. Nevertheless, additional studies, particularly in larger cohorts, are needed to fully understand its role and establish its effectiveness as a prognostic tool in the context of CV risk in RA.

## 6. Catestatin

CST is a bioactive peptide derived from the proteolysis of chromogranin A, and is widely expressed in various tissues and systems, including the CV, nervous, and immune systems. It functions as a multifunctional modulator of cellular homeostasis, playing a crucial role in the regulation of autonomic nervous system activity, catecholamine release, and inflammatory response [127] (Figure 4). Additionally, CST exerts anti-inflammatory, antioxidant, and cardioprotective effects, linking it to the prevention of endothelial dysfunction, vascular remodeling, and myocardial fibrosis. Its involvement in CVDs, such as heart failure, hypertension, and atherosclerosis, as well as in metabolic and inflammatory disorders, positions it as a promising biomarker and an emerging therapeutic target for the development of new diagnostic and treatment strategies.

### 6.1. Clinical Evidence on the Role of CST in CVD

CST plays a significant role in the regulation of CV functions, and its involvement in various CVDs is the focus of intense research. In patients with heart failure (HF), CST has shown dual behaviors and significant clinical associations. Higher CST levels have been detected in patients with acute decompensated HF, particularly in those with a history of AMI, and it has been identified as an independent predictor of in-hospital mortality in this context [128,129]. In chronic HF, recent studies have found that elevated CST levels increase the risk of all-cause mortality by up to fivefold, highlighting its utility as a prognostic marker in patients with HF with reduced ejection fraction [130]. On the other hand, in HF with preserved ejection fraction, CST appears to attenuate diastolic dysfunction by reducing the production of reactive oxygen species [131].

In the context of AMI, CST exhibits a transient pattern, with an increase in levels in the days following the event. Longitudinal studies have shown that elevated CST levels on day three post-AMI correlate with left ventricular remodeling, worse ventricular function, and poorer long-term outcomes [132,133,134]. The authors of these studies attribute these CST increases to a compensatory mechanism aimed at inhibiting catecholamine release during AMI to mitigate their harmful effects. Additionally, CST has been identified as an independent predictor of malignant arrhythmias in AMI patients, further reinforcing its clinical relevance in this setting [135]. This compensatory CST increase has also been observed in patients with atrial fibrillation (AF) [136], and animal models have shown that CST treatment reduces the risk of post-AMI AF [137].

In other CV diseases, studies have found lower CST levels in patients with coronary artery disease compared to healthy controls, with an inverse association between CST levels and atherosclerosis severity [138]. In patients with ACS, lower CST levels were observed compared to controls, although no direct relationship with prognosis was found [139]. However, in HF patients after physical exertion, CST levels predicted hospitalizations and all-cause mortality during a two-year follow-up [140].

CST also has important links to metabolic syndrome and CV risk factors. In patients with hypertension and T2DM, CST levels have been found to be significantly reduced, with correlations observed between CST and lipid profile, insulin resistance, and anthropometric parameters [141]. Similarly, low CST levels have been associated with essential hypertension [142]. These findings suggest a potential role in the pathophysiology of this condition. However, existing data indicate a more complex relationship. Some studies have reported higher CST concentrations in hypertensive patients compared to controls [143,144] and in untreated hypertensive patients compared to those receiving treatment, with CST levels correlating with pulse wave velocity [143,144], suggesting a possible link to arterial stiffness. Furthermore, previous research did not detect significant associations between CST levels and blood pressure [145,146], highlighting the need for a more nuanced interpretation. It has been proposed that, in the early stages of the disease, CST levels may be low, whereas in advanced stages, they may increase as a compensatory mechanism in response to sympathetic nervous system activation [143,144].

### 6.2. Clinical Evidence on the Role of CST in RA

To date, the only published evidence on the role of CST in rheumatoid arthritis comes from a study conducted in 2022. This research showed that CST levels were significantly higher in RA patients compared to a control group and were significantly associated with disease duration and RA inflammatory activity [147]. These initial findings suggest a potential role for CST in the pathophysiology of RA, although additional studies are needed to confirm these results and further explore its clinical relevance.

### 6.3. Mechanisms of Action of CST

CST plays a multifaceted role in CV and metabolic regulation, acting through various protective and modulatory mechanisms. It has been described that CST contributes to blood pressure reduction by promoting histamine release and nitric oxide production, as well as inhibiting catecholamine secretion. It also improves insulin sensitivity, attenuates obesity, and regulates hepatic and plasma lipid levels, suggesting its involvement in metabolic homeostasis. These effects may be further enhanced by the antioxidant activity of CST [142].

Additionally, CST exhibits an anti-atherogenic mechanism mediated by angiotensin-converting enzyme 2, which inhibits both the production of proinflammatory cytokines and leukocyte adhesion to endothelial cells, thereby contributing to vascular wall protection [138]. Moreover, CST has been shown to suppress macrophage-mediated atherogenesis by reducing inflammation in endothelial cells and macrophages, inhibiting foam cell formation, and regulating the balance between collagen-1, elastin, and fibronectin, promoting a more stable vascular structure [148]. In animal models, CST administration has been found to slow the progression of atherosclerosis, reinforcing its potential therapeutic role in this context [148].

Furthermore, in vitro models have demonstrated that CST promotes angiogenesis with an effect comparable to that of the vascular endothelial growth factor (VEGF) through a mechanism dependent on the basic fibroblast growth factor (bFGF) and has shown the ability to reduce cardiomyocyte apoptosis [149]. Collectively, its broad spectrum of actions suggests a significant potential as a future therapeutic tool for managing metabolic syndrome and CVDs.

### 6.4. Evidence Supporting CST as a Biomarker of CVD Risk in RA

Currently, there is no evidence supporting the use of CST as a prognostic biomarker for CV mortality in RA patients. Although CST has been associated with CV risk in other contexts, its specific role in RA has not been studied. In fact, the existing data are limited, originating from research on other chronic inflammatory diseases and based on surrogate risk markers. In this regard, a study published in 2020 showed that CST levels were significantly elevated in patients with inflammatory bowel disease compared to a control group. This study also detected an increased pulse wave velocity in the patient group, which correlated with CST levels, suggesting a potential relationship between this protein and arterial stiffness [150]. Additionally, a recent study found elevated CST levels in RA patients associated with rheumatoid factor and anti-citrullinated peptide antibodies without an observed association between CST levels and surrogate CV risk markers such as carotid intima-media thickness or the presence of carotid plaque [87]. These studies open the door for future research to explore the use of CST as a CV risk biomarker in RA.

## 7. Fetuin-A

Fet-A is a plasma glycoprotein primarily synthesized in the liver and secreted into circulation, where it regulates key processes such as inflammation, vascular calcification, and metabolism [151,152] (Figure 5). Its role in mineral homeostasis, insulin receptor signaling, NLRP3 inflammasome activation, and lipid mobilization in adipose tissue links it to CVD, chronic inflammatory disorders such as RA, and metabolic dysfunction. These functions connect chronic joint inflammation with an increased CV risk in RA, positioning Fet-A as a promising biomarker for risk stratification and a potential therapeutic target.

### 7.1. Clinical Evidence on the Role of Fet-A in CVD

Fet-A has been extensively studied regarding its association with CV mortality and the incidence of CV events. A 2013 study evaluating 753 patients with acute coronary syndrome found that low Fet-A levels combined with elevated C-reactive protein levels were associated with an increased risk of CV death [153]. These findings were supported by a meta-analysis published in 2018, which included data from four studies and a total of 4256 patients with coronary artery disease. This analysis confirmed that elevated serum Fet-A levels were associated with lower all-cause mortality in patients with coronary artery disease, emphasizing its potential as a protective biomarker in this context [154]. However, the relationship between Fet-A levels and CV outcomes appears to be influenced by the patient’s glycemic status. A study by Laughlin et al. in 2012 [155] revealed that the association between Fet-A and CV mortality varied depending on the presence or absence of T2DM. In non-diabetic individuals, low Fet-A levels were associated with a 76% increase in CV mortality risk, whereas in T2DM patients, low Fet-A levels were linked to a lower CV risk. These statistically significant associations were independent of traditional CV risk factors as well as renal and hepatic function. The authors suggested that the dual pathophysiological effect of Fet-A, preventing vascular calcification while promoting insulin resistance, could explain these findings in diabetic and non-diabetic subjects [155]. Similar results were reported in another 2013 study, where elevated Fet-A levels were associated with a lower CV risk in non-diabetic patients [156]. Subsequently, a 2016 study did not find a significant association between Fet-A levels and the overall CV risk. However, a trend toward a positive association in patients with T2DM or insulin resistance and an inverse trend in non-diabetic patients was observed, although these trends did not reach statistical significance [157].

Regarding subclinical CVD, one study analyzed the association between Fet-A levels and CVD in 1375 subjects without prevalent CVD, observing an inverse association between Fet-A levels and coronary artery calcification but not with blood pressure or cIMT measured by carotid ultrasound. The authors suggested that Fet-A may be involved in the vascular calcifications characteristic of advanced atherosclerosis but not in the early stages of disease development [158]. Furthermore, Fet-A has been linked to other CV risk factors, including T2DM pathophysiology [159] and metabolic syndrome [160]. It has also been observed that Fet-A levels can be modified by interventions such as weight loss and physical exercise, leading to a reduction in its plasma levels [161,162]. This suggests that Fet-A is not only a relevant biomarker but also a potential therapeutic target for CV risk modulation.

### 7.2. Clinical Evidence on the Role of Fet-A in RA

Fet-A levels in RA patients have been studied with conflicting results. Some research has reported reduced Fet-A levels in these patients. For instance, a 2007 study demonstrated that Fet-A levels in RA patients were significantly lower compared to a control group and were inversely correlated with inflammatory and nutritional markers. The authors suggested that this reduction could be related to a chronic inflammatory status and malnutrition associated with RA [163]. These findings were supported by a subsequent study in 2012, which also reported decreased Fet-A levels in RA patients [164]. However, other studies, particularly more recent ones, have reported elevated Fet-A levels in RA patients [165,166,167]. Unlike earlier studies, these investigations did not find a significant correlation between Fet-A levels and inflammatory markers such as the erythrocyte sedimentation rate or C-reactive protein, nor with disease activity indices [166,167]. A 2020 study highlighted that elevated Fet-A levels could predict a better response to rituximab treatment in RA patients who had previously failed anti-TNFα therapy, suggesting a potential role for Fet-A in therapeutic stratification [168]. In summary, although data are heterogeneous, Fet-A emerges as a biomarker with a multifaceted role in RA, involved in both inflammation modulation and treatment response.

### 7.3. Mechanisms of Action of Fet-A

Fet-A plays a dual role in the pathophysiology of RA and CVD, mediated by both anti-inflammatory and proinflammatory mechanisms. Recent studies suggest that Fet-A stabilizes extracellular calcium and phosphorus in calciprotein particles, which are internalized by monocytes, increasing lysosomal activity, activating the NLRP3 inflammasome, and promoting IL-1β release through calcium-sensing receptors, thereby amplifying joint inflammation [169]. Inhibiting the calciprotein uptake via calcium-sensing receptors has been proposed as a therapeutic approach to modulate inflammation in RA. Additionally, Fet-A has been shown to participate in the macrophage activation in RA, thereby regulating the activation of key immune cells involved in disease pathogenesis [170].

As a glycoprotein synthesized in hepatocytes, Fet-A exhibits anti-inflammatory effects by inhibiting the production of proinflammatory cytokines [171] and promoting the phagocytosis of apoptotic cells [172]. However, its role in metabolism is complex. The fatty liver/Fet-A pathway contributes to insulin resistance by inhibiting insulin receptor tyrosine kinase activity [173], underscoring its influence on metabolic disorders. Meanwhile, Fet-A acts as an arterial calcification inhibitor by increasing calcium and phosphorus solubility, preventing their deposition in the vascular wall [174,175,176]. This anti-calcification effect is crucial for the prevention of atherosclerosis, although in RA, mineral stabilization by Fet-A could paradoxically exacerbate inflammasome activation and inflammation [169,170]. Finally, a recent proteomic analysis confirmed the presence of Fet-A in atherosclerotic plaques, where it is retained without being actively expressed, suggesting its role in modulating advanced vascular calcification [177].

Taken together, these findings reveal different mechanisms of action of Fet-A that highlight its dual role as a multifunctional regulator of inflammatory and metabolic processes.

### 7.4. Evidence Supporting Fet-A as a Biomarker of CVD Risk in RA

Although there is evidence emphasizing the importance of Fet-A in RA pathophysiology, its association with atherosclerosis, and its role as an inflammatory and CV risk biomarker, no direct studies validate its utility as a prognostic biomarker for CV events in RA patients. In this regard, a recent study found no association between Fet-A levels and surrogate CV risk markers, such as cIMT or the presence of carotid plaque [87]. While Fet-A shows potential, its ability to predict specific CV events remains uncertain. These findings underscore the need for further research to explore its clinical applicability in CV risk assessment in this population.

## 8. Conclusions

Advances in the identification of emerging biomarkers have opened new perspectives in the assessment of CV risk in patients with RA. The biomarkers reviewed, including IL-32, DKK-1, Gal-3, CST, and Fet-A, have demonstrated involvement in the pathophysiology of CVD and the chronic inflammation that characterizes RA, with roles ranging from the modulation of inflammation and vascular remodeling to their involvement in arterial calcification and endothelial dysfunction (Table 1). The available evidence suggests that some of these biomarkers could improve the accuracy of CV risk assessment in RA, facilitating the identification of patients with a greater susceptibility to developing CVD. However, for others, the evidence remains very preliminary. Although some biomarkers, such as galectin-3, have attracted broader clinical interest, most remain at a preliminary stage of investigation in RA, and none are routinely used in clinical practice for CV risk assessment in this population. Moreover, no reference values are currently available, which complicates their immediate adoption in clinical settings. Most of the existing evidence comes from small or cross-sectional studies, highlighting the need for large-scale prospective trials to confirm these biomarkers’ predictive capacity for CV events in RA. Without such studies, incorporating them into clinical guidelines remains premature. Another challenge is that each biomarker may influence multiple pathways—such as inflammation, endothelial function, lipid metabolism, or vascular calcification—introducing possible confounding factors when interpreting results. Furthermore, there is a lack of evidence regarding the optimal frequency of measurement. Finally, the incremental value of these biomarkers over established risk factors (e.g., CRP and LDLc) remains unproven, and no studies have evaluated their cost-effectiveness for routine analyses, as most investigations thus far have been exploratory rather than focused on economic feasibility. Additionally, for all of them, significant knowledge gaps persist, preventing their routine clinical implementation. Well-designed prospective studies are needed to validate their utility as prognostic biomarkers for CV events in RA patients. In this regard, this review has highlighted that the identification of these new biomarkers in RA represents a significant advance in understanding the relationship between inflammation and CV risk. Not only could this improve risk stratification in these patients, but it could also facilitate the development of new therapeutic strategies to mitigate the impact of CV disease in RA, one of the leading causes of morbidity and mortality in this population.

## Figures and Tables

**Figure 1 biomedicines-13-00870-f001:**
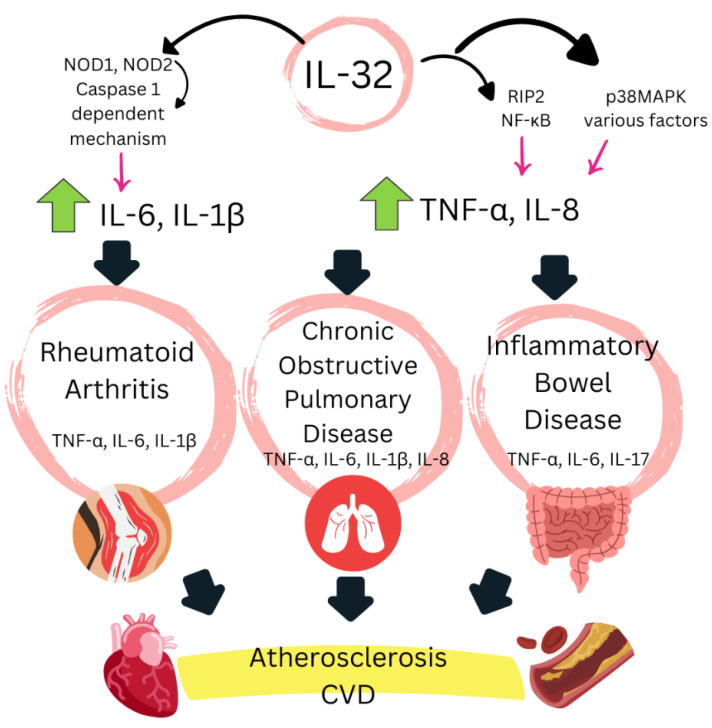
A schematic of the link between chronic inflammation (IL-32) and cardiovascular disease. IL-32 can act synergistically with NOD1 and NOD2 ligands, resulting in the production of IL-6 and IL-1β. Signaling through NOD2 can also lead to NF-κB activation via RIP2 signaling, leading to the production of TNFα and IL-8. IL-32, which alone can signal through p38 MAPK and NF-κB, triggering the production of TNFα and IL-8. This is adapted from [34]. TNF: tumor necrosis factor α; IL-32: interleukin-32; IL-6: interleukin-6; IL-1β: interleukin-1β; IL-17: interleukin-17; IL-8: interleukin-8; NOD: nucleotide-binding oligomerization domain proteins; NF-κB: nuclear factor kappa-light-chain-enhancer of activated B cells; RIP2: receptor-interacting protein 2; p38MAPK: p38 mitogen-activated protein kinases.

**Figure 2 biomedicines-13-00870-f002:**
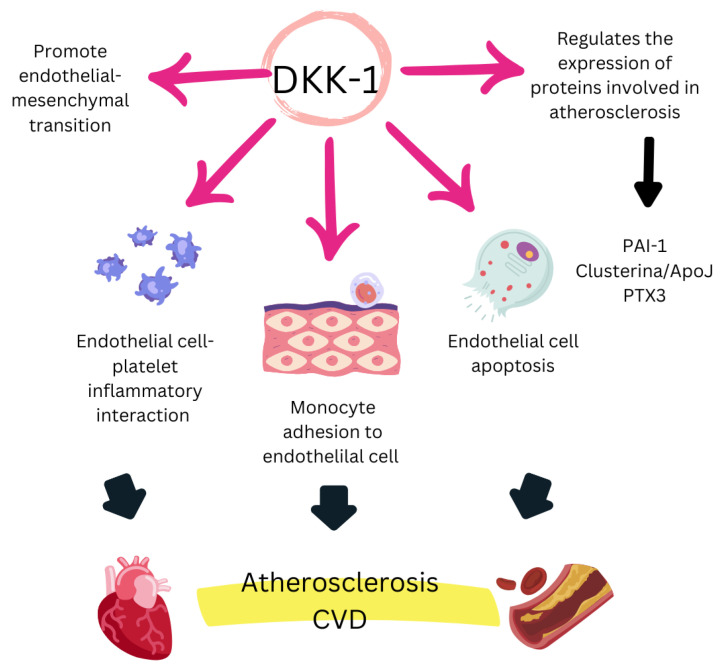
A schematic overview of the proatherogenic effects of Dkk-1, adapted from [57]. PAI-1: plasminogen activator inhibitor type 1; ApoJ: apolipoprotein J; PTX3: pentraxin 3.

**Figure 3 biomedicines-13-00870-f003:**
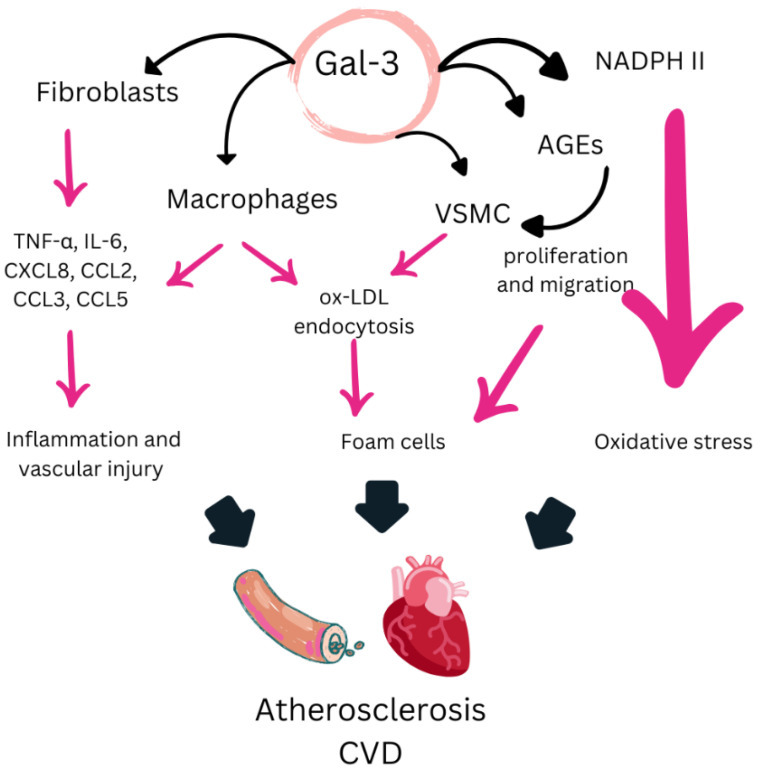
A diagram showing the mechanisms by which Gal-3 promotes the formation of atherosclerosis. Adapted from [89,90]. Gal-3: galectin-3; TNF-α: tumor necrosis factor-α; IL-6: interleukin-6; CXCL8: CXC motif chemokine 8; CCL2: CC chemokine ligand 2; CCL3: CC chemokine ligand 3; CCL5: CC chemokine ligand 5; ox-LDL: oxidized low-density lipoprotein; VSMC: vascular smooth muscle cells; AGEs: advanced glycation end products; NADPH II: nicotinamide-adenine dinucleotide phosphate II.

**Figure 4 biomedicines-13-00870-f004:**
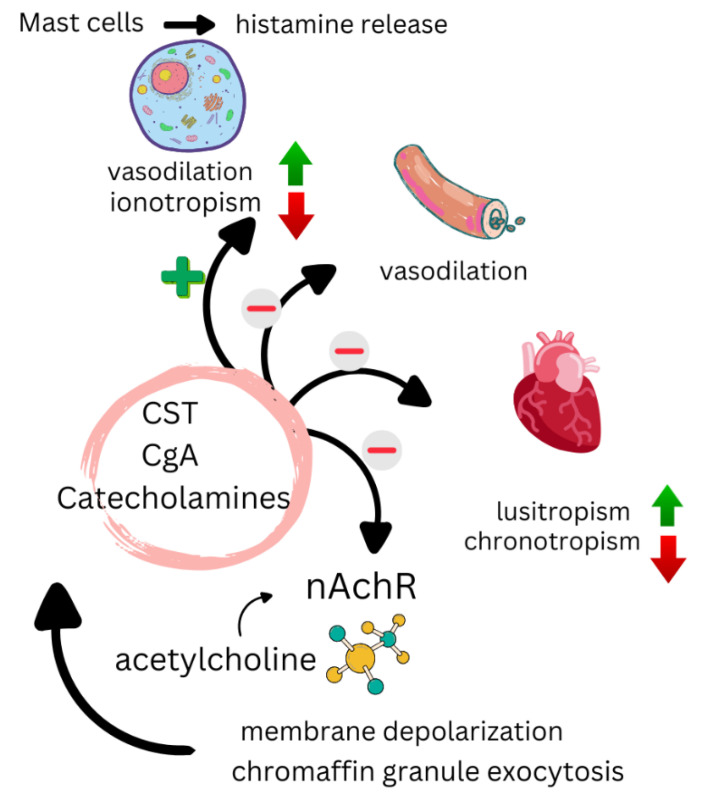
A diagram showing the functions of the CST and several of its receptors. The inhibition of nAchR interferes with membrane depolarization, which subsequently prevents the entry of the calcium necessary for the exocytosis of chromaffin granules, adapted from [127]. nAchR: nicotinic acetylcholine receptors; CST: catestatin; CgA: chromogranin A. Red arrow indicates decrease, green arrow indicates increase, + indicates activation, and − indicates inhibition.

**Figure 5 biomedicines-13-00870-f005:**
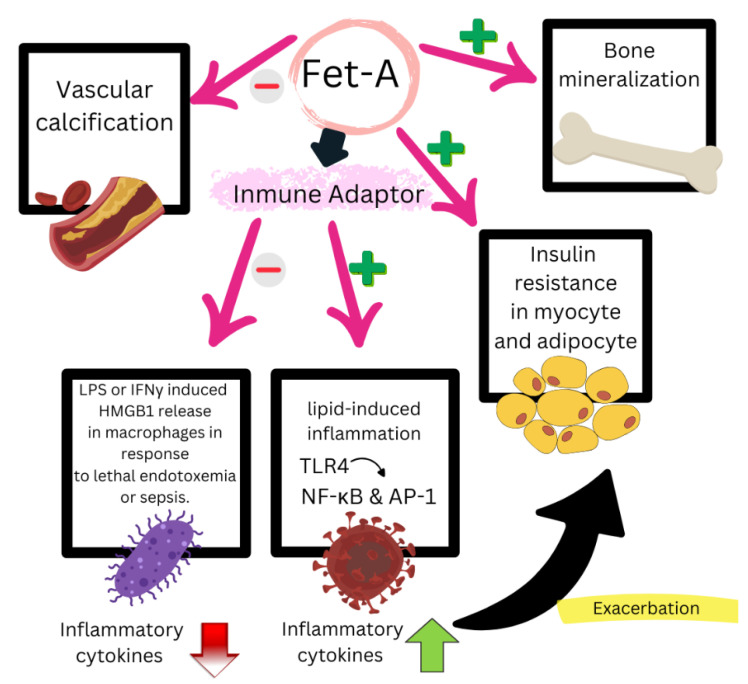
A diagram representing the role of fetuin-A. It improves the absorption and fixation of calcium and phosphate minerals. It acts as a transporter of insoluble phosphate and calcium, preventing the precipitation of calcium salts and the generation of vascular calcifications. It is a negative regulator of the innate immune response and an adaptor between FFAs and TLR4 signaling in lipid-induced inflammation, and TLR4 signaling leads to the activation of NF-κB and AP-1, adapted from [151,152]. HMGB1: High Mobility Group Box 1; LPS: lipopolysaccharide; IFN γ: interferon γ; FFAs: free fatty acids; TLR4: Toll-like receptor 4; NF-κB: nuclear factor kappa-light-chain-enhancer of activated B cells; AP-1: activating protein 1. + Indicates activation, − indicates inhibition, red arrow indicates decrease, and green arrow indicates increase.

**Table 1 biomedicines-13-00870-t001:** A summary of key roles, evidence in RA, the strength of clinical evidence, limitations, and current clinical use for the different biomarkers (IL-32, DKK-1, Gal-3, and CST, Fet-A). IL-32: interleukin-32; DKK-1: Dickkopf-1; Gal-3: galectin-3; CST: catestatin; Fet-A: fetuin-A; RA: rheumatoid arthritis; CVD: cardiovascular disease; cIMT: carotid intima-media thickness; cPP: carotid plaque presence.

Biomarker	ccKey Role	Evidence in RA	Clinical Evidence Strength	Limitations Current Clinical Use
IL-32	- Proinflammatory cytokine [33].	- Links synovial inflammation to endothelial dysfunction [34]. - Elevated in RA synovial fluid; correlates with inflammation [39,42]. - No association with cIMT or cPP [87].	- Moderate (cross-sectional studies, genetic associations). - Potential for risk stratification.	- Experimental. - Implicated in other chronic diseases and may not be exclusive to RA.
DKK-1	- Inhibits Wnt pathway; promotes vascular calcification. - Promotes atherosclerosis [58,59].	- Elevated in RA patients [76], highly expressed in synovial fluid and synovial fibroblasts [78]. - Associated with synovial fibroblast migration and inflammation [77]. - Associated with myocardial perfusion defects [85]. - No association with cIMT or cPP [87].	- Early-stage research. - Strong (meta-analyses, Mendelian randomization).	- Limited prognostic validation. - Early-stage research. - Experimental.
Gal-3	- Mediates fibrosis and inflammation and plaque instability [89].	- Elevated in serum and synovial fluid correlating with C-reactive protein and joint destruction [109]. - Correlated with anti-citrullinated peptide antibodies and joint erosions [111]. - Possible diagnostic biomarker [115]. - Decreases cardiac output and increases systemic vascular resistance [124]. - Links to aortic inflammation but not coronary calcification [126]. - No association with myocardial perfusion [125]. - No association with cIMT or cPP [87].	- Emerging in heart failure. - Moderate (conflicting data in RA cohorts).	- Inconsistent links to coronary calcification in RA. - Approved for HF prognosis (non-RA).
CST	- Regulates vascular calcification [127]. - Anti-inflammatory, cardioprotective [131].	- Higher in RA patients and associated with disease duration and disease activity [147]. - Associated with rheumatoid factor and anti-citrullinated protein antibody positivity in an RA cohort. No association with cIMT or cPP [87].	- Experimental. - Limited (single RA study).	- Limited evidence. - Experimental.
Fet-A	- Inhibits vascular calcification. - Regulates calcification; dual pro/anti-inflammatory roles.	- Inversely associated with the erythrocyte sedimentation rate in an overall RA cohort. No association with cIMT or cPP [87].	- Weak (heterogeneous RA data).	- Requires prognostic validation. - Experimental. - Conflicting associations with RA and CVD.

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
