# Peer review of "New Cardiovascular Risk Biomarkers in Rheumatoid Arthritis: Implications and Clinical Utility—A Narrative Review"

_biomedicines, 2025, doi:10.3390/biomedicines13040870_

Round 1
Reviewer 1 Report
Comments and Suggestions for Authors
Strengths:
- The article is well-structured, easy to follow. I appreciate the logical, constant and coherent progression.
- The data are up to date and supported by recent studies.
- The topic introduces new biomarkers that could improve risk stratification.
- The discussion includes potential clinical applications of the biomarkers, guiding future research.
However, there are a few areas that could be improved:
- Introduce a concise discussion on the biomarkers limitations and challenges in clinical practice. Although the article discusses potential clinical applications, it does not specify whether these biomarkers are already used in clinical practice or remain experimental.
- I think a more critical assessment is needed for some of the biomarkers analyze; for example you stated that IL-32 is associated with cardiovascular risk, but it is unclear whether it is specific to RA or merely a general inflammatory marker.
- At some points, there are inconsistent data; for example, not all studies confirm a strong relationship between galectin-3 and cardiovascular risk in patients with RA; I think a discussion detailing the factors that might influence these variations would add value to the article.
- A section/a table comparing these biomarkers to those already established as cardiovascular risk factors (such as CRP, LDL-C); such comparison would add clinical relevance and show where these biomarkers provide added value. For example, you can provide a table including: biomarkers, advantages, limitations and if it is already used in clinical practice.
- A summary table that compares all the described biomarkers would be welcomed; it clarifies clinical applicability; improves readability.
Refine writing for clarity and conciseness!
Reduce excessive text and redundancy!
Some sentences are overly complex and difficult to follow. For example:
- page 2 --> "In this shared pathogenic process, an increase in the synthesis of proinflammatory...";
- page 2 --> "Furthermore, chronic inflammation in RA reduces...";
- page 15 --> “The available evidence suggests that…”;
- in the abstract à “Identifying and incorporating these biomarkers into routine clinical practice could lead to better management of cardiovascular risk in RA patients, thus improving outcomes in this high-risk population”. “thus improving outcomes in this high-risk population” à is redundant; if management improves, outcomes will too.
Author Response
Comments and Suggestions for Authors.
- Introduce a concise discussion on the biomarkers limitations and challenges in clinical practice. Although the article discusses potential clinical applications, it does not specify whether these biomarkers are already used in clinical practice or remain experimental.
Response: Thank you for the comment as it highlights an important aspect. We have included the following discussion (page 18, lines 691-706) addressing the main limitations and challenges inherent to these emerging biomarkers.
“Although some biomarkers, such as galectin-3, have attracted broader clinical interest, most remain at a preliminary stage of investigation in RA, and none are routinely used in clinical practice for CV risk assessment in this population. Moreover, no reference values are currently available, which complicates their immediate adoption in clinical settings. Most of the existing evidence comes from small or cross-sectional studies, highlighting the need for large-scale prospective trials to confirm these biomarkers’ predictive capacity for CV events in RA. Without such studies, incorporating them into clinical guidelines remains premature. Another challenge is that each biomarker may influence multiple pathways—such as inflammation, endothelial function, lipid metabolism, or vascular calcification—introducing possible confounding factors when interpreting results. Furthermore, there is a lack of evidence regarding the optimal frequency of measurement. Finally, the incremental value of these biomarkers over established risk factors (e.g., CRP, LDLc) remains unproven, and no studies have evaluated their cost-effectiveness for routine analyses, as most investigations thus far have been exploratory rather than focused on economic feasibility.”
- I think a more critical assessment is needed for some of the biomarkers analyze; for example, you stated that IL-32 is associated with cardiovascular risk, but it is unclear whether it is specific to RA or merely a general inflammatory marker.
Response: We appreciate the suggestion. In the revised manuscript, we have expanded our explanation of IL-32 by emphasizing two key points: its association with conditions beyond RA and its potential disease-specific implications for RA (pages 5-6, lines 207-227)
“Furthermore, in patients with non-alcoholic fatty liver disease and metabolic dysfunction, IL-32 levels have been associated with poor blood pressure control, the primary CV risk factor (53). However, its association with CV risk extends beyond RA. Elevated IL-32 levels have also been observed in other chronic inflammatory conditions, such as metabolic disorders and HIV infection, which indicates that this cytokine may not be exclusive to RA. For instance, in patients with HIV, increased IL-32 levels have been associated with subclinical markers of atherosclerosis (54–56). Additionally, IL-32 has been implicated in other chronic diseases with increased CV risk, such as chronic obstructive pulmonary disease (COPD) and inflammatory bowel disease (34), reinforcing its broader role in systemic inflammation. Nevertheless, its overexpression in synovial tissue—particularly the IL-32γ isoform—suggests a distinct disease-specific dimension in RA. IL-32’s interplay with TNF-α and other inflammatory mediators is known to drive synovial inflammation and endothelial dysfunction, both of which could further heighten CV risk in this patient population. Future studies should focus on longitudinal evaluations that compare IL-32 levels in RA with those in other inflammatory disorders to determine whether its predictive value is indeed specific to RA-related CV complications or reflects a more generalized inflammatory state.
Overall, the available evidence suggests that IL-32 may serve as a candidate biomarker for CV risk in RA, supported by its involvement in RA pathophysiology, atherosclerosis, and systemic inflammation. In addition, its role in disease mechanisms makes it a promising therapeutic target worthy of further investigation.”
- At some points, there are inconsistent data; for example, not all studies confirm a strong relationship between galectin-3 and cardiovascular risk in patients with RA; I think a discussion detailing the factors that might influence these variations would add value to the article.
Response: We appreciate the reviewer’s comment regarding the variability in the data concerning Gal-3 and its relationship with CV risk in RA. We have added the following text to discuss this point (page 11-12, lines 459-464):
Variations in the reported relationship between Gal-3 and CV risk in patients with RA may be partly explained by its differential utility across disease stages. Gal-3 could be more informative in the early stages of atherosclerosis, when vascular changes are potentially reversible, while its predictive value may decline in more advanced stages of CVD. Additionally, differences in study design, patient selection, and assay methodologies likely contribute to the observed inconsistencies.
- A section/a tablecomparing these biomarkers to those already established as cardiovascular risk factors (such as CRP, LDL-C); such comparison would add clinical relevance and show where these biomarkers provide added value. For example, you can provide a table including biomarkers, advantages, limitations and if it is already used in clinical practice.
Response: We appreciate the reviewer’s suggestion. While we agree that such a comparison could add clinical context, a comprehensive table comparing all biomarkers in depth extends beyond the scope of our current manuscript. Instead, we have acknowledged this as a limitation in our Discussion (page 18, lines 702-706) with the following statement.
Finally, the incremental value of these biomarkers over established risk factors (e.g., CRP, LDLc) remains unproven, and no studies have evaluated their cost-effectiveness for routine analyses, as most investigations thus far have been exploratory rather than focused on economic feasibility.”
- A summarytable that compares all the described biomarkers would be welcomed; it clarifies clinical applicability; improves readability.
Response: Thank you for your interesting suggestion. We have included a summary table (page 17) comparing all biomarkers, detailing their key roles, evidence in RA, clinical evidence strength, limitations, and current clinical use. We believe this table improves readability, highlights the preliminary nature of the evidence, and provides a more critical perspective by categorizing the level of evidence for each biomarker.
Reviewer 2 Report
Comments and Suggestions for Authors
- The organization of the manuscript requires enhancement. There is an abrupt transition from the introduction to the section on biomarkers, which lacks appropriate subheadings. This deficiency contributes to a decrease in the overall readability of the document.
- While this is a narrative review, it is recommended that the authors include a methods section that outlines the keywords and search engines utilized in their research.
- While the authors primarily concentrate on "Implications and Clinical Utility," it may enhance the manuscript's value if they also briefly consider the issue of cost-effectiveness. Additionally, Figure 3 has the potential to be more educational and informative; therefore, I recommend that the authors provide further elaboration on this figure.
- The authors indicate that existing evidence implies that certain biomarkers may enhance the precision of cardiovascular (CV) risk assessment in individuals with rheumatoid arthritis (RA), thereby aiding in the identification of patients who are at a heightened risk of developing cardiovascular disease (CVD). However, for several biomarkers, the evidence remains largely preliminary. Furthermore, significant knowledge gaps exist for all biomarkers, which hinder their routine application in clinical practice. To establish their efficacy as prognostic indicators for CV events in RA patients, well-structured prospective studies are essential. I recommend that the authors consider publishing a separate paper that specifically addresses the biomarkers requiring further investigation. This could illuminate the existing gaps in the literature and provide guidance for future research directions.
- In conclusion, it may be beneficial to develop a simplified diagram that encompasses all relevant biomarkers that could improve the accuracy of cardiovascular (CV) risk assessment in patients with rheumatoid arthritis. This diagram could be presented as a graphical abstract. The final decision regarding this matter is left to the authors.
Author Response
Comments and Suggestions for Authors
- The organization of the manuscript requires enhancement. There is an abrupt transition from the introduction to the section on biomarkers, which lacks appropriate subheadings. This deficiency contributes to a decrease in the overall readability of the document.
Response: We appreciate the reviewer’s comment. We would like to clarify that the structure of the biomarker sections was already designed with four consistent subheadings to improve readability and ensure a logical presentation of the information. Each biomarker is discussed under the following standardized subheadings:
-
- Clinical Evidence on the Role of [Biomarker] in CVD
- Clinical Evidence on the Role of [Biomarker] in RA
- Mechanisms of Action
- Evidence Supporting [Biomarker] as a Biomarker of CVD Risk in RA
This structure was implemented to facilitate comprehension and to allow direct comparisons between biomarkers.
Additionally, in response to reviewer feedback and as requested by other reviewers, we have incorporated a Methodology section (page 3 line 110-121) between the Introduction and the Biomarker Results. We believe this addition further improves the overall flow of the manuscript.
- While this is a narrative review, it is recommended that the authors include a methods section that outlines the keywords and search engines utilized in their research.
Response: Thank you for this suggestion. We have introduced a Methods section in the manuscript to detail the literature search strategy, including the databases used, keywords, and timeframe covered. Below is the revised text we have included in the manuscript (page 3 line 110-121).
Methods:
For this narrative review, we conducted a comprehensive search of the PubMed, MEDLINE, and ScienceDirect databases. Our focus centered on five biomarkers (IL-32, DKK-1, Gal-3, CST, and Fet-A) and their associations with CV disease and CV risk in RA. The search strategy involved using each biomarker’s name as a primary keyword in combination with “CV disease” and “CV risk”, as well as additional searches combining each biomarker’s name with “rheumatoid arthritis” to ensure specificity to RA-related CV risk. To further refine the results, we conducted additional searches by cross-referencing biomarker-related cardiovascular terms with “rheumatoid arthritis”. The search was restricted to articles published from the year 2000 onward. An initial screening identified 1,908 potentially relevant records, of which 177 met our inclusion criteria following abstract review.
- While the authors primarily concentrate on "Implications and Clinical Utility," it may enhance the manuscript's value if they also briefly consider the issue of cost-effectiveness.
Response: Thank you very much for this interesting suggestion. We agree that cost-effectiveness is an important aspect when evaluating the clinical utility of biomarkers. However, since these biomarkers are currently in the research phase and have not yet been implemented in routine clinical practice, no cost-effectiveness studies are available. Given the importance of this aspect, we have acknowledged it as a limitation in the Discussion (page 18, line 704-706) with the following statement:
"… and no studies have evaluated their cost-effectiveness for routine analyses, as most investigations thus far have been exploratory rather than focused on economic feasibility."
- Additionally, Figure 3 has the potential to be more educational and informative; therefore, I recommend that the authors provide further elaboration on this figure.
Response: Thank you for your recommendation. We have redesigned and expanded Figure 3. Additionally, we have included new figures for each biomarker to further enhance clarity and comprehension.
- The authors indicate that existing evidence implies that certain biomarkers may enhance the precision of cardiovascular (CV) risk assessment in individuals with rheumatoid arthritis (RA), thereby aiding in the identification of patients who are at a heightened risk of developing cardiovascular disease (CVD). However, for several biomarkers, the evidence remains largely preliminary. Furthermore, significant knowledge gaps exist for all biomarkers, which hinder their routine application in clinical practice. To establish their efficacy as prognostic indicators for CV events in RA patients, well-structured prospective studies are essential. I recommend that the authors consider publishing a separate paper that specifically addresses the biomarkers requiring further investigation. This could illuminate the existing gaps in the literature and provide guidance for future research directions.
Response: Thank you for your comment. We agree that the evidence for several biomarkers remains preliminary, and we have acknowledged this by adding a paragraph in the Discussion addressing the existing knowledge gaps and challenges in their clinical application (page 18, lines 691-706). Additionally, we have included a summary table that categorizes the level of evidence for each biomarker alongside other relevant information to provide a clearer perspective.
We also appreciate your suggestion to publish a separate paper focusing on the biomarkers that require further investigation, and we might take this recommendation into consideration as we further develop our research in this area.
- In conclusion, it may be beneficial to develop a simplified diagram that encompasses all relevant biomarkers that could improve the accuracy of cardiovascular (CV) risk assessment in patients with rheumatoid arthritis. This diagram could be presented as a graphical abstract. The final decision regarding this matter is left to the authors.
Response: Thank you for your suggestion. To improve the clarity of our review and to incorporate feedback from all reviewers, we have decided to include a comparative table summarizing all biomarkers (page 17). This addition will provide a clearer overview of the biomarkers and facilitate a better understanding of their roles and effects. Additionally, we have added new figures for each biomarker to further enhance readability and comprehension. We believe these modifications improve the overall structure of the manuscript and provide a more informative synthesis of the available evidence.
Reviewer 3 Report
Comments and Suggestions for Authors
The authors conducted a narrative review of biomarkers IL-32, DKK-1, Galectin-3, Catestatin (CST), and Fetuin-A (Fet-A) to improve CVD risk assessment in RA patients. Although the high risk of CVD in RA patients is well documented, studies in the literature that emphasize the clinical applicability of these biomarkers are limited.
Title & Abstract: The title clearly reflects the content of the study. The abstract summarizes the scope of the study well, but more specific clinical recommendations would be useful.
Introduction: It summarizes the association of RA with CVD well and emphasizes the importance of the topic, but is unnecessarily long. In addition, it would be useful to consider the different effects of CVD risk factors in RA in a more systematic way.
Main Biomarkers: Detailed mechanisms, clinical evidence, and associations with RA are discussed for each biomarker. The biomarkers discussed in the study have different levels of clinical applicability. There is more evidence for IL-32 and DKK-1 in particular, while evidence for Catestatin and Fetuin-A is more limited. The study could provide a more critical perspective by indicating the levels of evidence for each biomarker. In addition, more comments could be added on how the biomarkers under review can be integrated into clinical practice. For example, which markers can be used together with standard biomarkers in CVD risk assessment? There is a mechanism diagram for Galectin-3, but if similar visuals were added for other biomarkers, its clarity would be increased. In addition, it would be great to have a table summarizing the studies conducted in the context of RA for each biomarker.
Methodology: Since the study is a narrative review, not a systematic review, the data selection criteria were not clearly stated. Providing more information on the literature search method, selection criteria and quality of the included studies could increase the methodological strength of the study. More details should be provided on the literature selection (e.g., which databases were used?).
Discussion and Conclusion: The potential of CVD biomarkers is highlighted, but clear recommendations for clinical practice and further directions for translational research could be added.
Author Response
Comments and Suggestions for Authors
The authors conducted a narrative review of biomarkers IL-32, DKK-1, Galectin-3, Catestatin (CST), and Fetuin-A (Fet-A) to improve CVD risk assessment in RA patients. Although the high risk of CVD in RA patients is well documented, studies in the literature that emphasize the clinical applicability of these biomarkers are limited.
- Title & Abstract: The title clearly reflects the content of the study. The abstract summarizes the scope of the study well, but more specific clinical recommendations would be useful.
Response: Thank you for your recommendation. We have added the following sentence to the abstract (page 1 line 37) to provide more specific clinical context.
Currently, these biomarkers are in the research phase and are not yet implemented in standard care.
- Introduction: It summarizes the association of RA with CVD well and emphasizes the importance of the topic but is unnecessarily long. In addition, it would be useful to consider the different effects of CVD risk factors in RA in a more systematic way.
Response: We agree. We have rewritten the introduction according to the reviewer’s comments (page 2-3).
- Main Biomarkers: Detailed mechanisms, clinical evidence, and associations with RA are discussed for each biomarker. The biomarkers discussed in the study have different levels of clinical applicability. There is more evidence for IL-32 and DKK-1 in particular, while evidence for Catestatin and Fetuin-A is more limited. The study could provide a more critical perspective by indicating the levels of evidence for each biomarker.
Response: Thank you for your suggestion. To address this comment, we have included a summary table (page 17) categorizing the level of evidence for each biomarker, highlighting differences in clinical applicability. This table provides a clearer perspective on the strength of the available data and further enhances the critical assessment of the study.
- In addition, more comments could be added on how the biomarkers under review can be integrated into clinical practice. For example, which markers can be used together with standard biomarkers in CVD risk assessment?
Response: Thank you for your comment. The integration of these biomarkers into clinical practice is an important aspect. However, they are currently in the research phase and have not yet been implemented in routine clinical practice. To address the reviewer comment, we have acknowledged this limitation in the Discussion section with the following statement (page 18, line 691-706), and we have also added a summary table that provides an overview of their clinical applicability alongside other relevant information.
“Although some biomarkers, such as galectin-3, have attracted broader clinical interest, most remain at a preliminary stage of investigation in RA, and none are routinely used in clinical practice for CV risk assessment in this population. Moreover, no reference values are currently available, which complicates their immediate adoption in clinical settings. Most of the existing evidence comes from small or cross-sectional studies, highlighting the need for large-scale prospective trials to confirm these biomarkers’ predictive capacity for CV events in RA. Without such studies, incorporating them into clinical guidelines remains premature. Another challenge is that each biomarker may influence multiple pathways—such as inflammation, endothelial function, lipid metabolism, or vascular calcification—introducing possible confounding factors when interpreting results. Furthermore, there is a lack of evidence regarding the optimal frequency of measurement. Finally, the incremental value of these biomarkers over established risk factors (e.g., CRP, LDLc) remains unproven, and no studies have evaluated their cost-effectiveness for routine analyses, as most investigations thus far have been exploratory rather than focused on economic feasibility.”
- There is a mechanism diagram for Galectin-3, but if similar visuals were added for other biomarkers, its clarity would be increased.
Response: We agree. To improve clarity, we have created new figures illustrating the mechanistic pathways for the biomarkers that previously did not have diagrams.
- In addition, it would be great to have a table summarizing the studies conducted in the context of RA for each biomarker.
Response: Thank you for your suggestion. To address this comment, we have included a summary table (page 17) that provides an overview of the key studies conducted in the context of RA for each biomarker. This table details their key roles, available evidence, clinical applicability, and limitations, helping to clarify the current state of research and highlight existing knowledge gaps. We believe this addition improves the readability and critical assessment of the study.
- Methodology: Since the study is a narrative review, not a systematic review, the data selection criteria were not clearly stated. Providing more information on the literature search method, selection criteria and quality of the included studies could increase the methodological strength of the study. More details should be provided on the literature selection (e.g., which databases were used?).
Response: Thank you for this suggestion. We have introduced a Methods section in the manuscript to detail the literature search strategy, including the databases used, keywords, and timeframe covered. Below is the revised text we have included in the manuscript (page 3 line 110-121)
Methods:
For this narrative review, we conducted a comprehensive search of the PubMed, MEDLINE, and ScienceDirect databases. Our focus centered on five biomarkers (IL-32, DKK-1, Gal-3, CST, and Fet-A) and their associations with CV disease and CV risk in RA. The search strategy involved using each biomarker’s name as a primary keyword in combination with “CV disease” and “CV risk”, as well as additional searches combining each biomarker’s name with “rheumatoid arthritis” to ensure specificity to RA-related CV risk. To further refine the results, we conducted additional searches by cross-referencing biomarker-related cardiovascular terms with “rheumatoid arthritis”. The search was restricted to articles published from the year 2000 onward. An initial screening identified 1,908 potentially relevant records, of which 177 met our inclusion criteria following abstract review.
- Discussion and Conclusion: The potential of CVD biomarkers is highlighted, but clear recommendations for clinical practice and further directions for translational research could be added.
Response: Thank you for your comment. The potential of CVD biomarkers is an important aspect, but their integration into clinical practice remains challenging due to the preliminary nature of the current evidence. To address this, we have acknowledged these limitations in the Discussion section (page 18, lines 691-706) and have added a summary table that provides an overview of their clinical applicability, highlighting the gaps that need to be addressed before their routine implementation. We believe that these additions enhance the manuscript’s discussion on the translational potential of these biomarkers while clarifying the current limitations that must be addressed through future research.
Round 2
Reviewer 3 Report
Comments and Suggestions for Authors
The revised version of the manuscript "New Cardiovascular Risk Biomarkers in Rheumatoid Arthritis: Implications and Clinical Utility. A Narrative Review" has been carefully reviewed. The authors have provided convincing responses to the previous comments, addressing the concerns effectively.
Additionally, the inclusion of Table 1 and the figures has significantly enhanced the quality of the manuscript, improving its clarity and comprehensiveness. These additions provide valuable support to the discussion and strengthen the overall presentation of the study.
I appreciate the authors’ efforts in revising the manuscript and believe that it is now much improved.